# Foliar application of fullerenol and zinc oxide nanoparticles improves stress resilience in drought-sensitive *Arabidopsis thaliana*

Ana Joksimović[1,2], Danijela Arsenov [3], Milan Borišev [3], Aleksandar Djordjević[2], Milan Župunski [4]*, Ivana Borišev [2]

1 A Bio Tech Lab Ltd, Sremska Kamenica, Serbia, 2 Department of Chemistry, Biochemistry and Environmental Protection, Faculty of Sciences, University of Novi Sad, Novi Sad, Serbia, 3 Department of Biology and Ecology, Faculty of Sciences, University of Novi Sad, Novi Sad, Serbia, 4 Institute of Cell and Interaction Biology, Heinrich Heine University, Düsseldorf, Germany

* milan.zupunski@hhu.de

## Abstract

This study aimed to characterise the chemical properties of fullerenol nanoparticles (FNP) and zinc oxide nanoparticles (ZnO nano), as well as their physiological and molecular effects following foliar application on *Arabidopsis thaliana*. Additionally, we explored the potential synergistic impact of combining ZnO nano with FNPs to enhance plant resilience under drought stress conditions. Chemical characterisation confirmed the successful formation of a stable FNP-ZnO aggregate. The previously established biostimulatory effects of fullerenol at micromolar concentrations were reaffirmed, highlighting its unique chemical properties. We demonstrated that a low dose (10 mg/L) of ZnO nano, used as a foliar application for the first time in *Arabidopsis thaliana*, positively influenced drought stress acclimatization. Our findings indicate that FNP and ZnO alleviate oxidative stress by mitigating the impact of reactive oxygen species (ROS), modulating antioxidant enzyme activities, and stabilising redox balance. Photosynthetic performance, stomatal conductance and water-use efficiency were optimized, particularly through fullerenol application, due to its unique antioxidative and hygroscopic properties. We further analyzed the expression of selected drought-response genes involved in ABA-dependent and ABA-independent water deficit acclimation in Col-0 wild-type and *pp2ca-1* drought hypersensitive mutant backgrounds. Our results revealed distinct gene expression changes in response to nanoparticle treatments, demonstrating modulation of ABA signaling and stress-related transcription factors. The combined application of FNP and ZnO exhibited unique, synergistic protective effects in drought acclimation. Future research will further elucidate the direct mechanisms linking these physiological outcomes to specific nanoparticle properties, paving the way for innovative strategies in sustainable agriculture.

**Data availability statement:** All relevant data are within the paper and its Supporting Information files.

**Funding:** The authors gratefully acknowledge the financial support of the Ministry of Science, Technological Development and Innovation of the Republic of Serbia (Grants No. 451 -03-137/2025-03/ 200125 & 451-03-136/2025-03/ 200125), and the DEAL initiative HHU (M.Ž.). The funders had no role in study design, data collection and analysis, decision to publish, or preparation of the manuscript.

**Competing interests:** I have read the journal's policy and the authors of this manuscript have the following competing interests: After initial submission of this manuscript, the authors became involved in a startup company focused on seed hydropriming using $C_{60}(OH)_{24}$ nanomaterials. This study investigates foliar application in Arabidopsis and was conducted entirely independent of the company, with no financial support or input from the startup. We disclose this for full transparency. This does not alter our adherence to PLOS ONE policies on sharing data and materials.

# Introduction

Drought is one of the most frequent environmental stressors massively affecting fundamental processes in plants, such as photosynthesis, transpiration rate, and stomatal conductance, consequently leading to changes in plant development, growth, and production and causing crop yield drop. Therefore, developing novel technologies to boost plant tolerance and resilience to drought stress is a critical global priority for food security. Such advancements hold the potential to significantly improve agricultural productivity by generating resilient crops with increased yield and quality. Recent studies revealed that nanotechnology is an emerging platform for producing "stress-smart plants" that can successfully confront current and future climate challenges [1]. NPs exhibit specific physicochemical properties, including a high surface area-to-volume ratio, unique chemical reactivity, thermal stability, surface charge, and electrical conductivity. Engineered nanoparticles (ENPs) are purpose-designed nanomaterials with at least one dimension below 100 nm, exhibiting tailored chemical and physical properties. Composed of metals, polymers, semiconductors, or carbon-based materials, ENPs are widely used in biomedicine, electronics, environmental and agricultural applications. These characteristics make ENPs suitable for advancing modern agriculture by offering tools to enhance crop growth and resilience [2]. The interaction of ENPs with soil and plants is primarily governed by their absorption capacity, behavior in soil matrices, stability in soil environments, uptake through roots and leaves, translocation through xylem and phloem, and interaction with key biomolecules such as proteins, RNA, and DNA [3]. Thus, these properties are crucial for the effective application and functionality of ENPs in agricultural settings [4].

Within the broader field of nanomaterials, carbon-based (in the first line, graphene, carbon nanotubes, and fullerenes) and metal-based nanoparticles (including nano ZnO) have garnered particular attention in recent years. These studies increasingly focus on the beneficial application of carbon-based nanoparticles due to their promising roles in enhancing plant health, productivity, and resilience, as reviewed by Sigala-Aquilar et al. 2024 [5].

Fullerenols (fullerenol nanoparticles, FNPs) are polyhydroxylated derivatives of fullerenes, including $C_{60}$, $C_{70}$, $C_{82}$, and others, with varying numbers of hydroxyl groups per molecule, influencing the unique physical, chemical, and biological properties of fullerenols [6]. Due to its commercial availability and good solubility in water, fullerenol with 24 hydroxyl groups [$C_{60}(OH)_{24}$] has shown the most promising reproducibility among all synthesized fullerenols, making it a benchmark in the biological field [7]. The synthesized FNP used in this study contains 24 hydroxyl groups – a form with which we have extensive experience across various *in vitro* and *in vivo* biological models. Fullerenols bearing 18 or 40 hydroxyl groups differ in both structural organization and surface charge characteristics compared to the 24-hydroxylated FNP, which can ultimately result in distinct biological responses. This is related to the synthesis procedure that yields the purest single product with a defined number of hydroxyl groups and with stable values of surface charges at a wide pH range [7,8].

Throughout the last decade, numerous studies have addressed the impact of hydroxylated fullerenes on plants, with most findings pointing to concentration-dependent beneficial effects that vary depending on treatment duration and methods, including foliar, seed, and root applications. Many studies suggest that fullerenols hold promise as a sustainable and effective tool for enhancing plant growth and resilience to various environmental stresses by triggering various biochemical, morphological, physiological, and molecular modifications in different species. For instance, fullerenol promoted hypocotyl elongation in *Arabidopsis thaliana* [9] and root growth in barley [10] and wheat [11]. FNP can also promote biomass production and water content [12], essential nutrient uptake [13,14], improve the efficiency of photosynthesis [15], modulate secondary metabolite levels, and alleviate oxidative stress by serving as a reactive oxygen species (ROS) sponge [16–18]. Recent advancements, including our previous study [16] highlighted the potent role of FNPs against drought stress. Xiong and coworkers [15] stress that foliar fullerenol applications induced abscisic acid (ABA) biosynthesis in drought-stress *Brassica napus* by down-regulating ABA catabolic genes, indicating hormone-related immune response. Further, transcriptomic analysis revealed changes in gene expression related to carbohydrate and amino acid metabolism and secondary metabolite synthesis, which improved water deficit tolerance [19]. The great potential of water-soluble hyper-harmonized hydroxyl-modified fullerene in addressing water scarcity is provided by Subotić et al. (2022), who reported a reduction of oxidative stress through upregulation of aquaporin gene expression in drought-stressed *Solanum lycopersicum* plants [18].

Besides FNPs, metal-based nanoformulations, such as zinc oxide nanoparticles, are frequently used NPs to mitigate abiotic stresses. The rationale for ZnO nanoparticles usage relies on the fact that Zn is an essential micronutrient with a pivotal role in plant growth and development. Therefore, foliar application of micronutrients is increasingly favoured in field applications due to its environmentally friendly approach. It offers advantages over traditional soil fertilisation, often posing challenges such as toxicity or reduced accessibility from chelation with soil organic matter [20]. Recent research has focused on establishing effective concentrations and application protocols for ZnO nanoparticles to support plant growth and enhance stress tolerance, aiming to position these nanomaterials as viable tools for sustainable agriculture. Pullagurala et al. (2018) identified a concentration threshold, suggesting that low doses of ZnO nanoparticles, less than 20 mg/l in foliar applications, tend to enhance plant growth and specific stress tolerance traits [21]. Recent findings reviewed this evidence by emphasising the promotive effect of foliar application of Zn nanoparticles in photosynthetic activity, modulating enzymatic processes and related gene expression, enhanced nutrient and carbon use efficiency, and altered secondary metabolites supporting antioxidant defenses [22,23].

The other side of the coin is the potential toxicity of NPs, which is mainly linked to NPs concentration and duration of plant exposure to nanomaterials. Research by Kovel et al. (2021) suggests that fullerenols with fewer -OH groups tend to exhibit higher toxicity than those with more hydroxyl groups, indicating a functional relationship between hydroxylation and biological effects [24]. Conversely, the toxicity of ZnO nanoparticles is mainly attributed to their high intracellular concentrations, leading to excessive ROS production [25–28]. Despite the significant struggles, it is still unclear how NPs behave in the plant-soil system and how they trigger plants' defence mechanisms, including metabolomic, proteomic, and genomic strategies, to cope with various environmental stresses. Even though drought stress responses in the model organism *Arabidopsis thaliana* have been extensively explored, sufficient knowledge of NP behaviour and their impact on the biological systems is still lacking [29]. For instance, it is well known that ABA is a core component that orchestrates many adaptive responses in plants, including activation/inhibition of stress-responsive genes [30]. However, insufficient comprehensive research on the underlying mechanisms that lead to enhanced drought tolerance is still missing since different species and ecotypes possess variations in plant plasticity to cope with harsh environmental conditions [30]. Therefore, using model organisms and different ecotypes can be a benchmark for understanding pathways and networks involved in drought response. Also, comprehension of their role at the ultrastructural and molecular levels is added value in defining their function, whether NPs act as stress indicators or stress inhibitors [1]. Furthermore, understanding the mechanisms behind NP-plant interaction is pivotal before adopting them in agricultural systems.

Given all these insights, one of our research objectives is to examine the effects of low-dose foliar applications of ZnO nanoparticles (10 mg/l) on drought stress resistance in *Arabidopsis thaliana*, specifically by assessing physiological responses and drought-related gene expression. To the best of our knowledge, the impact of low-dose ZnO nanoparticle foliar treatments on *A. thaliana* has not been previously explored. We further hypothesise that a combined application of ZnO nanoparticles and fullerenol nanoparticles (FNPs) could integrate the unique properties of each, creating a synergistic formulation with potential agricultural applicability. Lastly, we propose that unique fullerenol nanoparticle properties can be eventually observed through physiological and metabolic profiling of foliar-treated plants, focusing on the expression of selected drought-responsive genes in drought-stressed *Arabidopsis thaliana*.

## Materials and methods

### Chemicals used

The chemicals used were sourced from Merck: HCl (115186); $Br_2$ (207888), ethanol (100974); NaOH (221465); $C_{60}$ (379646); $ZnSO_4 \cdot 7H_2O$ (221376); bulk ZnO (108849); nano ZnO (544906). The fullerenol containing 24 hydroxyl groups ($C_{60}(OH)_{24}$) was synthesized through a two-step chemical process. In the first step, catalytic bromination of $C_{60}$ with elemental bromine was carried out in the presence of $FeBr_3$. This reaction yields a single reaction product, $C_{60}Br_{24}$, without occluded bromine, and with well-defined *Th* symmetry [31]. In the second step, the bromine atoms were fully substituted by hydroxyl groups through a reaction carried out in an alkaline aqueous environment. The resulting reaction product is a molecule exhibiting high symmetry with a defined arrangement of –OH groups [8]. In aqueous solution, the fullerol molecules form stable nanoparticles with a zeta potential of approximately −25 mV to −50 mV [16,31,66].

### Nanoparticles characterization

Dynamic light scattering (DLS) was used to determine the hydrodynamic size of nanoparticles in water, while electrophoretic light scattering (ELS) was used to measure surface charge (zeta potential, $\zeta$). The first samples were measured after prior sonication, simultaneously with foliar application (treatment time). A second set of DLS and $\zeta$ potential analyses of the samples were conducted to assess whether 24-hour standing resulted in changes in particle size distribution and sample electrification. Measurements were performed using a Zetasizer Nano ZS (Malvern Instruments Inc, UK). All DLS analyses were conducted in triplicate, while zeta potential measurements were performed in duplicate. Powder morphology was characterised by scanning electron microscopy (SEM) using an Apreo 2 C Scanning Electron Microscope, Thermo Fisher Scientific, Waltham, MA, USA. Powder samples were gold-coated under a high vacuum prior to measurement. X-ray diffraction (XRD) was measured using a Rigaku Miniflex 600 Powder X-ray diffractometer. FTIR spectra of the samples were recorded on a Thermo Nicolet iS20 FTIR spectrophotometer (Thermo Fisher Scientific, Waltham, MA, USA) with Smart iTR™ ATR Sampling accessories, in the range of 4000–400 $cm^{-1}$.

### Plant material and experimental design

*Arabidopsis thaliana* plants were grown in pots with ready-to-use soil, with seeds sown immediately after sterilization. The plants were cultivated under controlled growth conditions of 14/10 h light/dark cycle, day/night temperatures of 22°C/18°C, with illumination intensity of 100 µmol/m²/s. Two genotypes were used in this study: (1) ecotype Columbia (Col-0) wild type and (2) *pp2ca-1* insertional mutant, encoding the Protein Phosphatase 2 C(PP2C), a negative regulator of the abscisic acid signalling pathway [32]. Since PP2Ca is a negative regulator of the abscisic acid pathway, plants typically exhibit a more pronounced response to drought stress when this gene is disrupted. The plants were grown for approximately 25 days before foliar treatments were applied. The treatments included: Control (distilled water), FNP100 (100 µmol/L of fullerenol), FNP100-ZnO nano (100 µmol/L of fullerenol with 10 mg/L ZnO nanoparticles), ZnO nano (10 mg/L of ZnO nanoparticles), and bulk ZnO (mock Zn treatment with 10 mg/L of bulk ZnO). After treatment, irrigation was withheld from

half of the plants to induce drought stress, culminating eight days later when soil moisture content reached 8–10 vol%. Ten plants per treatment were analysed, with 5–7 plants used for photographic documentation (supplemental material, S1 Fig in S1 File). Photosynthetic parameters were measured in vivo, and plants were subsequently harvested for biochemical and molecular analyses.

## Photosynthetic parameters

Photosynthetic rates (PN) and stomatal conductance (gs) were measured using an LCpro+ Portable Photosynthesis System (ADC BioScientific Ltd), with 12 replicates measured across four plants per treatment. Water use efficiency (WUE) was calculated as the ratio of photosynthetic rate to transpiration rate (PN/T), while intrinsic water use efficiency (iWUE) was calculated as the ratio of photosynthetic rate to stomatal conductance (PN/gs). All measurements were taken under controlled light conditions with photosynthetically active radiation (PAR) set at 800 $\mu mol\ m^{-2}\ s^{-1}$ using the LCpro+ light unit. A consistent ambient airflow of 100 $\mu mol\ s^{-1}$ was maintained, with constant temperature, humidity, and $CO_2$ levels throughout the experiments.

Relative chlorophyll content (SPAD), leaf thickness, and chlorophyll fluorescence parameters were measured using the MultispeQ V 2.0 device (PhotosynQ Inc., USA) following the Photosynthesis RIDES 2.0 protocol. Eight leaves per treatment were analyzed, and measured parameters were: gH+ (thylakoid membrane conductivity to protons), vH+ (steady-state proton flux), Phi2 (quantum yield of photosystem II), Fv/Fm (variable-to-maximal fluorescence ratio), PhiNQ (non-regulatory energy dissipation), and NPQt (non-photochemical quenching).

## Biochemical stress markers and antioxidant activity analyses

Four technical replicates were pooled from 10 plants per treatment for each biochemical parameter. Protein concentration was determined using the Bradford method [33], with results expressed in mg of protein per gram of fresh plant material. Reduced glutathione (GSH) content was quantified following the protocol by Kapetanović and Mieyal, (1979) [34]. Plant material was homogenised in 5% sulfosalicylic acid and centrifuged at 3000 rpm for 10 minutes. The supernatant was mixed with Ellman's reagent, and absorbance was measured at 412 nm after a 5-minute reaction. GSH content was expressed as $\mu mol$ of reduced glutathione per mg of protein. Proline content was measured following Bates et al. (1973) [35] with modifications by Lee et al. (2018) [36], where plant material was homogenised in sulfosalicylic acid, centrifuged, and the supernatant mixed with a 1.25% ninhydrin solution in 80% acetic acid. The mixtures were incubated at 100°C for 30 minutes, and absorbance was measured at 595 nm. Proline content was expressed as nmol per gram of fresh plant mass. Lipid peroxidation (LP) was assessed by measuring thiobarbituric acid reactive substances (TBARS) extracted from fresh plant material using a mixture of thiobarbituric acid (TBA), 10% perchloric acid (PCA), and 20% trichloroacetic acid (TCA) in a 1:3 ratio. Test solutions consisted of 0.25 ml of plant extract and 2.25 ml of TBARS extraction solution. After incubation at 95°C for 30 minutes, followed by cooling and centrifugation (3000 rpm for 10 minutes), the absorbance of the supernatant was measured at 532 nm. TBARS content was expressed as nmol equivalents per mg of protein [37].

Ascorbate peroxidase (APx, E.C. 1.11.1.11) activity was determined spectrophotometrically by monitoring the reduction in absorbance at 290 nm upon $H_2O_2$ addition, following, Nakano and Asada (1981) [38] and Amako et al. (1994) [39]. Catalase (CAT, EC 1.11.1.6) activity was measured following the absorbance decrease at 240 nm [40].

## RNA Isolation and quantitative RT-PCR analysis

Total RNA was isolated from 0.1 g of fresh plant material (leaves) from each treatment, frozen in liquid nitrogen, and stored at −80°C until analysis. RNA isolation was performed using the Qiagen RNeasy Plant Mini Kit (catalogue number 74903). After isolation, RNA concentration in the samples was measured using a Biospec Nano Shimadzu UV spectrophotometer (Shimadzu, Biotech). According to the manufacturer's instructions, total RNA was reversely transcribed into cDNA using the High-Capacity RNA-to-cDNA Kit (catalogue number 4387406 AB). The reaction was run in an Eppendorf Mastercycler

Gradient PCR machine (model 5331) with the following thermal program: 37°C for 60 minutes, 95°C for 5 minutes, and cooling at 4°C. Further, cDNA concentration was determined using the Qubit 2.0 Fluorometer (Invitrogen-Thermo Scientific, USA), with satisfactory results of <20 ng/20 µl reaction and absorbance ratios OD 260/280 > 1.7 and OD 260/230 > 2.2.

Gene expression analysis was performed using all ten samples and the following oligonucleotide primers targeting specific genes: AREB1/ABF, PYL4, PP2CA, PYR1, ABF3, RD29B, ABI2, DREB19, DREB2A, DREB2B, RD29a, along with reference genes ACT2 (S1 Table in S1 File). According to a pre-established protocol, the reactions were carried out on a QuantStudio 5 Real-Time PCR system (Applied Biosystems, Thermo Fisher Scientific). Reaction preparation was conducted under sterile conditions in a laminar flow cabinet (BioBase BSC-1300 II A2-X).

The experiment type was ΔΔCt, with the Melt Curve option included. The reaction was set to fast run mode with the following thermal cycling program: initial step 95°C for 20 s, denaturation 95°C for 1 s, annealing 60°C for 20 s, repeated for 40 cycles, followed by the melt curve stage: 95°C for 1 s, 60°C for 20 s, and 95°C for 1 s. The calibrator used was control (optimal conditions), and the endogenous control was the reference gene ACT2. Results of relative gene expression (RQ) are presented as log10 values.

## Data analysis

Data were analyzed, tested, and plotted with R version 4.1.2 and R Studio version 2022.02.3 (R Core Team 2021) and presented as mean ± standard error (SE). We employed a mixed-effects model with Type II ANOVA to analyze data in general, but also data characterized by tied values in the control group, with two-factor variables alongside one random variable. This approach was chosen to appropriately account for the nested structure of the data, where observations within each group are not independent, and to robustly test the main effects and interactions of the factors while controlling for other predictors in the model. Type II ANOVA was preferred as it provides unbiased estimates of main effects and interactions without being influenced by the order of terms in the model, ensuring accurate inference despite the presence of tied values and potential complexity in the experimental design. The mixed-effects model was fitted using the lmerTest package [41] in R, with the model formula y~treatment * condition + (1|rep), accommodating the nested structure of the gene expression data with tied values in the control group and two-factor variables alongside one random variable. The ANOVA analysis "car" package [42] was conducted to assess the significance of the main effects and interactions. Estimated marginal means were derived using the emmeans package [43], followed by pairwise comparisons with Tukey adjustment and letter-based comparisons using the cld function with Sidak adjustment for multiple comparisons, providing comprehensive insights into the differential expression patterns across treatment conditions. Ionome content data were analyzed using a linear model with the formula y~treatment × condition, followed by a two-way ANOVA (Type I) for each parameter to test the effects of treatment, condition, and their interaction. Post hoc pairwise comparisons (Sidak-adjusted) were conducted for all treatment × condition combinations to identify specific group differences for each analyzed element. Detailed ANOVA output tables for each parameter are provided in the Supplementary Information (S2–S5 Tables in S1 File). Data visualisations were done with "ggplot2" package [44].

Principal Component Analysis (PCA) was performed with the ade4 package [45] on correlation matrices. Evaluation of components retained in PCA was done with "paran" package [46], which employs Horn's parallel analysis [47] to check for finite sample bias in the retention of components. The approach contrasted the eigenvalues from PCA performed on random datasets (30 × (dataset variables)) with the same as in the input dataset. Together with inspecting PCA biplots, we decided to keep the first two principal components (PC) for further analysis and interpretation. Package factoextra was used for plotting PCA biplots [48].

## Results and discussion

### Physico-chemical characterization

To explore the relationship between the chemical properties of nanoparticles and the observed physiological effects in treated plant models, we first conducted a detailed characterization of the nanoparticles used. To get better insight into the

structure and to determine the nanoparticles' physical and chemical properties, the samples were analyzed using Scanning electron Microscopy (Fig 1), Dynamic Light Scattering (DLS) and zeta potential measurements (Fig 2), X-ray diffraction (Figs 3 and 4), and FTIR analyses (Fig 5).

SEM images of nano ZnO nano (Fig 1b, c) reveal the surface and characteristic crystal shape of ZnO nanoparticles measuring 49.06 nm. SEM micrograph of FNP (Fig 1a) shows the formation of amorphous, irregular rod-shaped submicron particles, consistent with images from previous studies [49]. The FNP/ZnO nano combination under 20000x magnification (Fig 1d) shows a significantly different morphology than the FNP micrographs in Fig 1a. Under the same magnification of 20000x, characteristic micrometre-scale rod-shaped forms of FNP cannot be observed (unpublished results), nor can the surface of ZnO nano. Particles formed in water from ZnO nano and FNP are better visible under 100000x and 250000x magnification, measuring 31,89 nm.

Dynamic Light Scattering (DLS) measurements of aqueous solutions of nano FNP100 revealed particle size distributions by number, with a peak at 38 nm representing 21%, consistent with previous results6 (Fig 2a, red line). The measured ζ potential of the FNP100 solution was −25.5 mV (Fig 2b, red line). Particle size distribution by number of ZnO nano showed that most of the particles have a size of about 295 nm (Fig 2a, black line), implicating that ZnO nano particles in water form agglomerates since separate individual particles have sizes of about 20–40 nm (Fig 3, XRD of ZnO nano). The surface charge (ζ potential) of ZnO nano was measured and expressed as a mean value of −21.5 mV (Fig 2b, blue line). In the combination of FNP100 and ZnO nano at a molar ratio of FNP 100 µmol/ZnO nano 1.3 µmol, the particle size distribution by number showed that two sizes of about 91 nm (21%) and about 220 nm (24%) (Fig 2a, blue and green line,

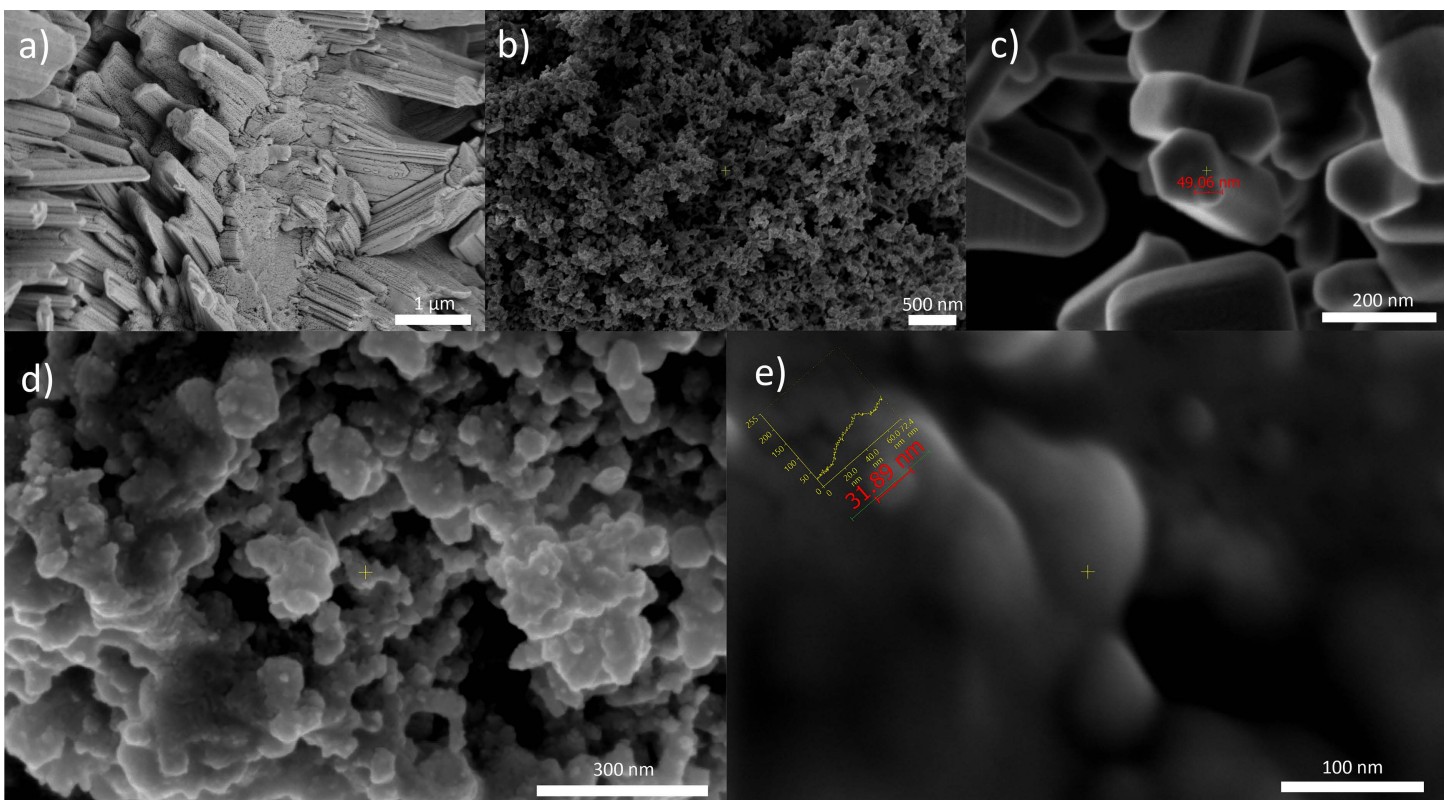

**Fig 1. SEM images of a) FNP at a magnification of 20000x; b,c) nano ZnO of 25000x and 150000x, respectively; d, e) SEM images of FNP100-ZnO nano at a molar ratio of FNP 100 µmol/ZnO nano 1.3 µmol at magnifications of 100000x, and 250000x, respectively.**

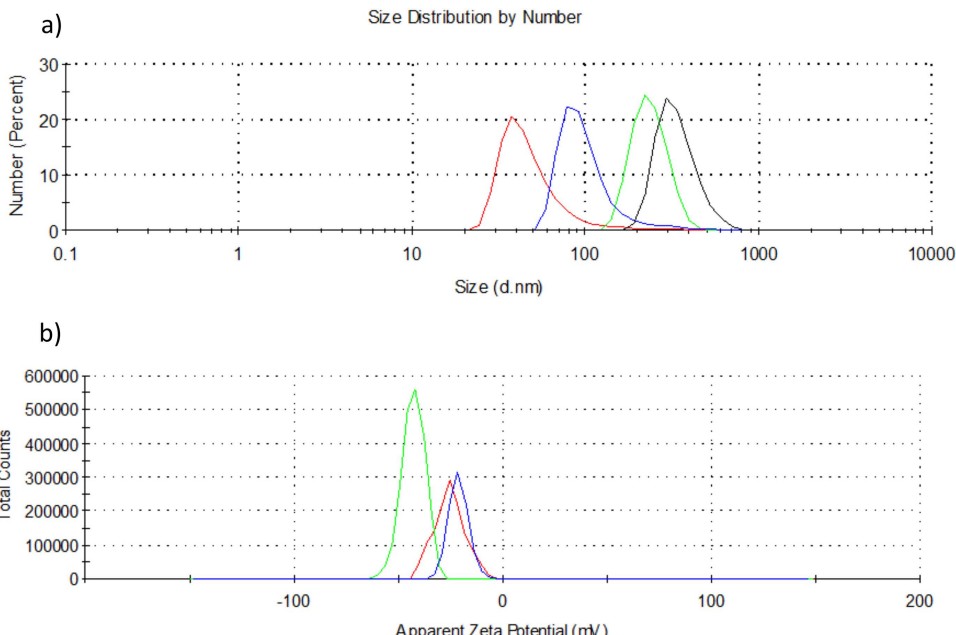

**Fig 2. DLS and zeta potential analyses.** Particle size distributions by number: a) FNP100 (red line), ZnO nano (black line) and FNP100-ZnO nano combination at a molar ratio of FNP 100 µmol/nano ZnO 1.3 µmol, after 30 minutes, in aqueous solution (green line, ZnO as referent material and blue line, FNP as referent material). b) ζ potential of FNP100 (red line), ZnO nano (blue line) and FNP100-ZnO nano combination at a molar ratio of FNP 100 µmol/nano ZnO 1.3 µmol (green line), in aqueous solution.

respectively) highlighted as dominant, due to the material used as a referent. The size of the particles of the FNP100-ZnO nano combination, 91 nm, can be considered the result of mutual repulsion of negative surface charges, indicating that FNP inhibits the self-aggregation of ZnO nano particles in the aqueous solution.

After 24 hours of standing in the dark, the solutions were remeasured, with no change observed except with the FNP100-ZnO nano combination sample, where the particle size distribution was slightly shifted towards 164 nm for both referent materials used (S3a Fig in S1 File), indicating the stabilisation of the combination agglomerates. These consecutive measurements indicate slower formation of stable FNP100-ZnO nano composite nanoparticles, confirming the correlation with SEM measurements. The ζ potential of FNP100-ZnO nano at a molar ratio of FNP 100 µmol/nano ZnO 1.3 µmol was about −42 mV regardless of the material used as referent (Fig 2b, green line). After 24h, the mean value of zeta potential shifted towards −35 mV, confirming the consolidation of the FNP100-ZnO nano combination (Supplement, S3b Fig in S1 File). The obtained surface charge values of the FNP100-ZnO nano formulation indicate the formation of stable nanoparticles in water at a pH of around 7.

Fig 3 shows the XRD diffractograms of nano and bulk ZnO. Fig 4 represents the diffractograms of the FNP and FNP/ZnO nano combination (FNP100-ZnO nano) in a molar ratio of FNP 100 µmol/ZnO nano 1.3 µmol, along with peaks for determining the crystal size.

X-ray structural analysis of ZnO nano powder yielded similar crystallite size results from three peaks. Analysis of the three peaks (Fig 3a–3c) yielded similar crystallite size results for nano and bulk ZnO samples: peak 1 ZnO nano (blue peak) 23 nm; bulk ZnO (red peak) 34 nm, peak 2 ZnO nano (blue peak) 22 nm.

Based on the FTIR spectra shown in Fig 5, the characteristic bands for ZnO nano observed at 3447 cm$^{-1}$, 2923 cm$^{-1}$, 1635 cm$^{-1}$, 1384 cm$^{-1}$, 1066 cm$^{-1}$, 701 cm$^{-1}$, and 510 cm$^{-1}$, are consistent with literature reports [50–52]. The band at 510 cm$^{-1}$ corresponds to Zn–O stretching vibrations, while the broad peak at 3447 cm$^{-1}$ indicates O–H stretching

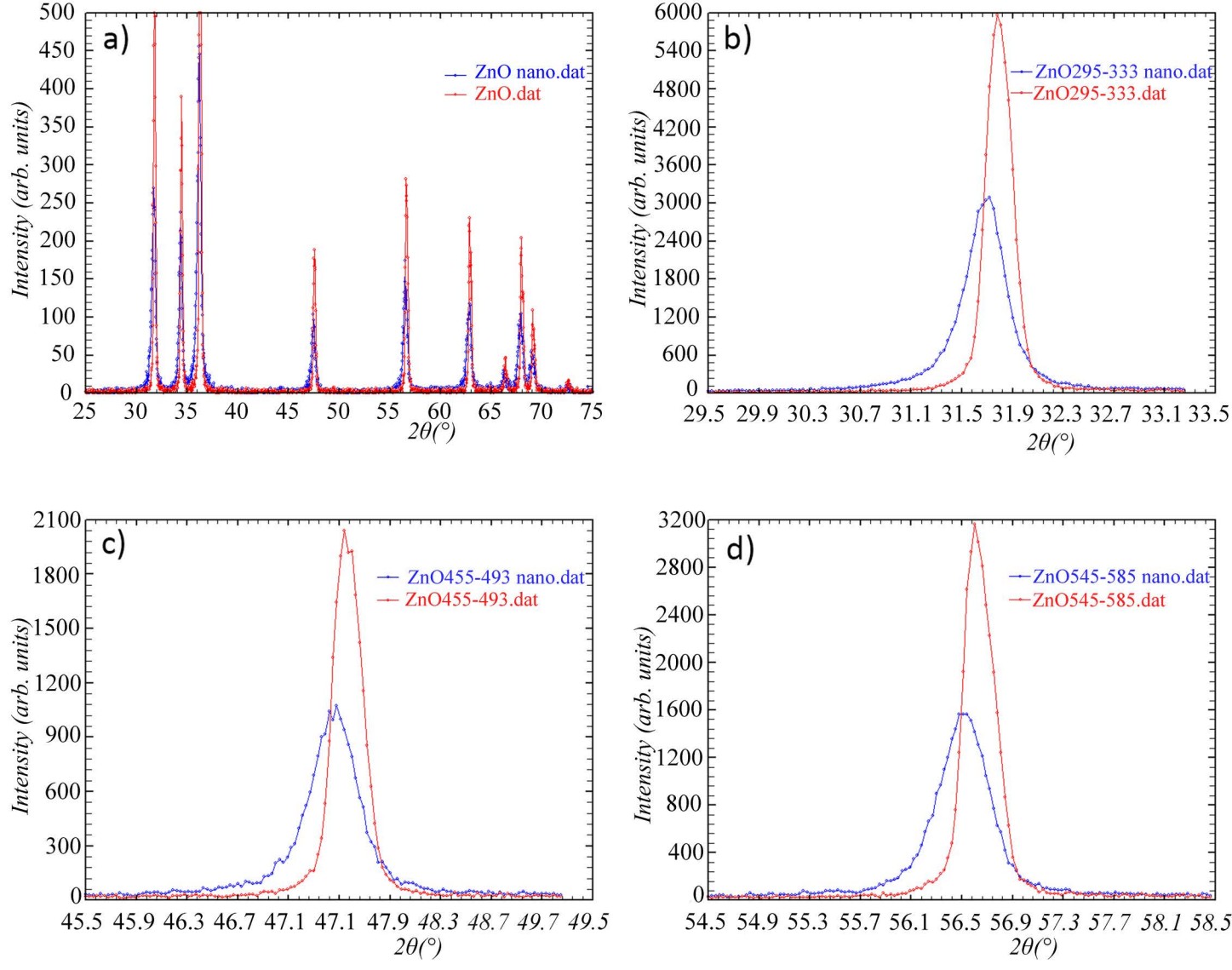

**Fig 3. XRD diffractograms of ZnO nano and bulk ZnO and crystallite sizes determined from three peaks: a) Diffractogram of ZnO nano and bulk ZnO; b) Peak 1 at 31.7° (ZnO nano – blue peak) NP 23 nm; (bulk ZnO – red peak) NP 34 nm; c) Peak 2 at 47.5° (ZnO nano – blue peak) NP 22 nm; (bulk ZnO – red peak) NP 32 nm; d) Peak 3 at 56.4° (ZnO nano – blue peak) NP 20 nm; (bulk ZnO – red peak) NP 31 nm.**

vibrations. Additional bands at 2923 cm$^{-1}$ and 1635 cm$^{-1}$ reflect bending vibrations, highlighting the presence of hydroxyl groups due to atmospheric moisture [53]. For FNP, FTIR spectra show characteristic bands at 3405 cm$^{-1}$ (O–H stretching), 1595 cm$^{-1}$, 1384 cm$^{-1}$ (C=C and C–C stretching), 1694 cm$^{-1}$, and 1075 cm$^{-1}$ (C–O stretching), which align with reported findings [54,55]. The FTIR spectrum of the FNP and ZnO nano combination exhibits shifts and new bands compared to individual components. The results of FTIR measurements indicate that the shift of the OH band from 3405 cm$^{-1}$ (in FNP) and 3447 cm$^{-1}$ (in ZnO nano) to 3440 cm$^{-1}$ in the combination suggests the formation of hydrogen bonds between the OH groups in FNP and ZnO nano (Fig 5). The band at 1595 cm$^{-1}$ (C=C) in FNP shifts to 1622 cm$^{-1}$, and the band at 1076 cm$^{-1}$ (C-O) shifts to 1097 cm$^{-1}$, indicating the presence of compelling electrostatic interactions between ZnO nano and FNP. The

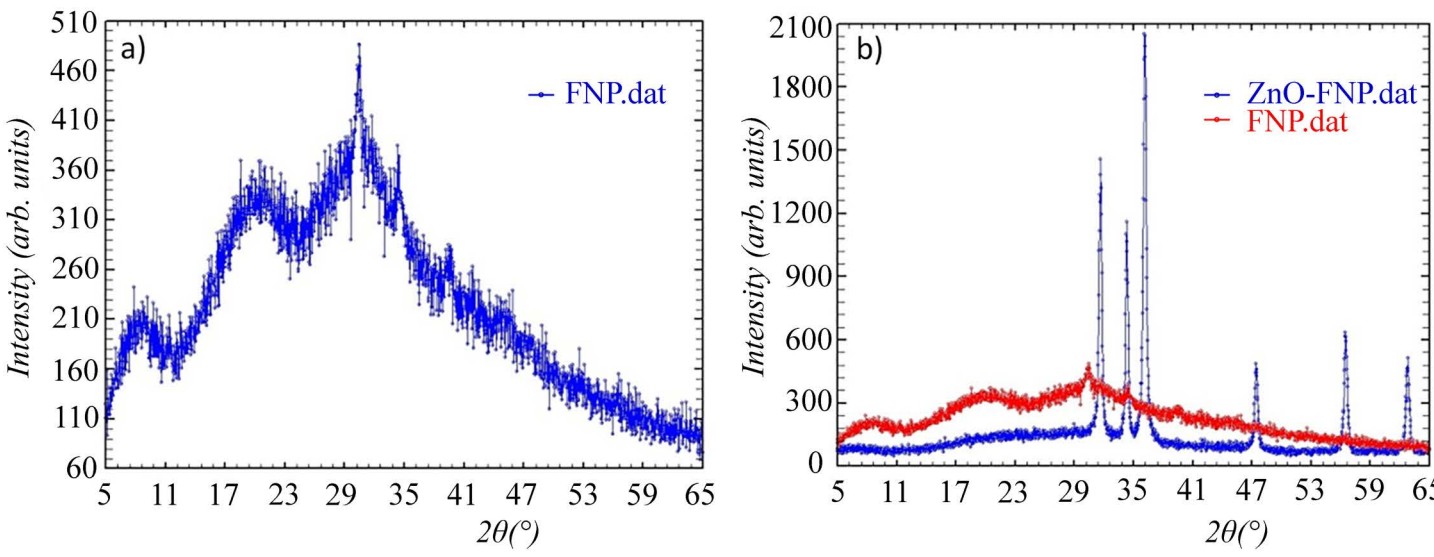

**Fig 4. XRD diffractograms of a) FNP and b) FNP and FNP/ZnO nano combination.**

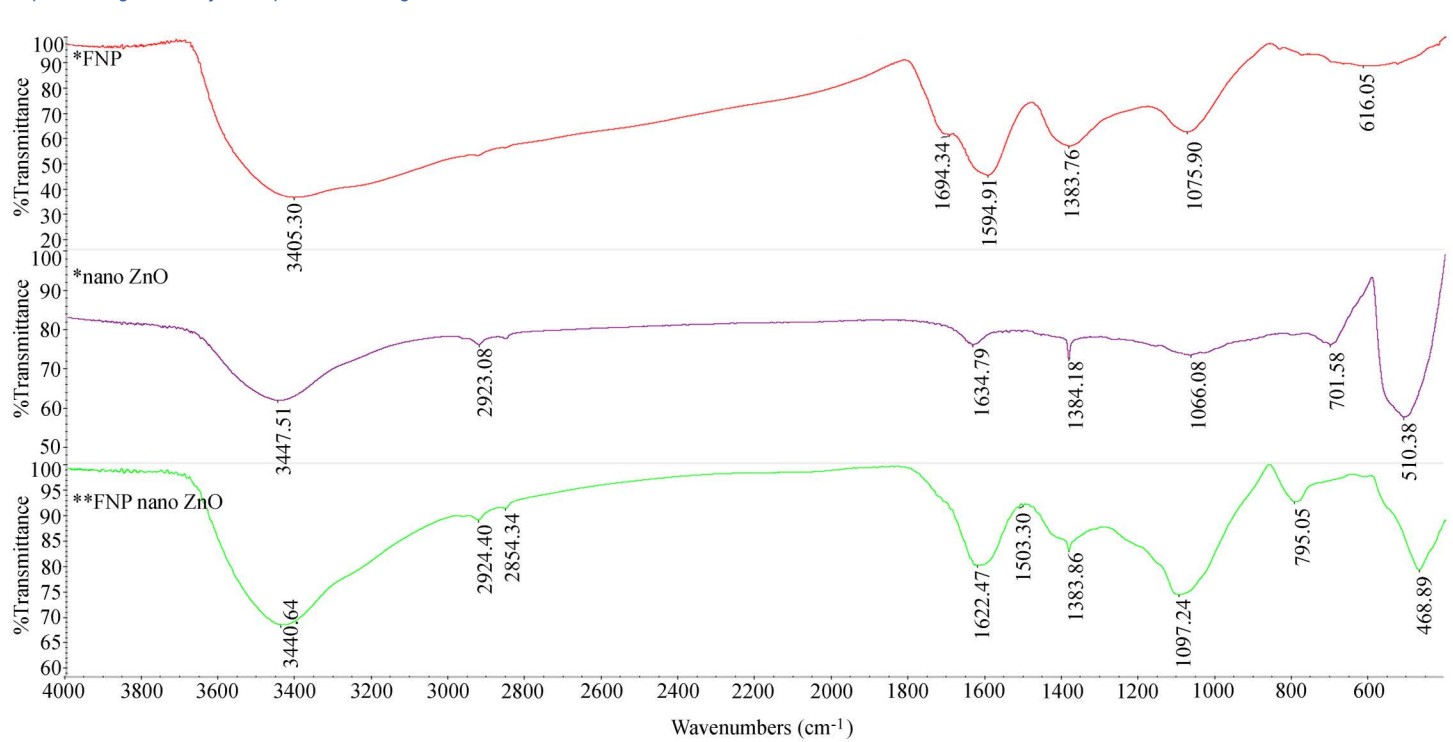

**Fig 5. FTIR analysis of the FNP, ZnO nano, and FNP100-ZnO nano.**

appearance of the C-O band (1097 cm$^{-1}$) in the combination indicates the preservation of these groups and interactions with ZnO nano. New bands at 795 cm$^{-1}$ and 469 cm$^{-1}$, likely arising from M–O bonds, reinforce the structural interaction between FNP and ZnO nano, while some ZnO nano bands at 701 cm$^{-1}$ and 510 cm$^{-1}$ disappear. FTIR analysis supports the conclusion that FNP and ZnO nano particles are potentially interconnected through hydrogen bonding.

## Assessment of leaf phenotypic changes indicates NP specificities

Treatments application minimally affected the leaf thickness, a differential of leaf mass and area (Fig 6). However, under control conditions in the Col-0 background, NP treatments barely affected this parameter. In contrast, treatment with bulk ZnO led to an almost 50% increase in thickness. This observation is interesting since it might imply differential utilization of zinc in two applied forms. Also, it putatively shows that the effect of NP is channeled through other means of activity, minimally affecting morphology. Analysis of leaf thickness in the *pp2ca-1* background revealed a reduction under FNP100 treatment in both optimal and drought conditions (Fig 6). However, the changes were not significantly affected. Leaf rosette phenotypes are provided in the supplemental information (S1 Fig in S1 File).

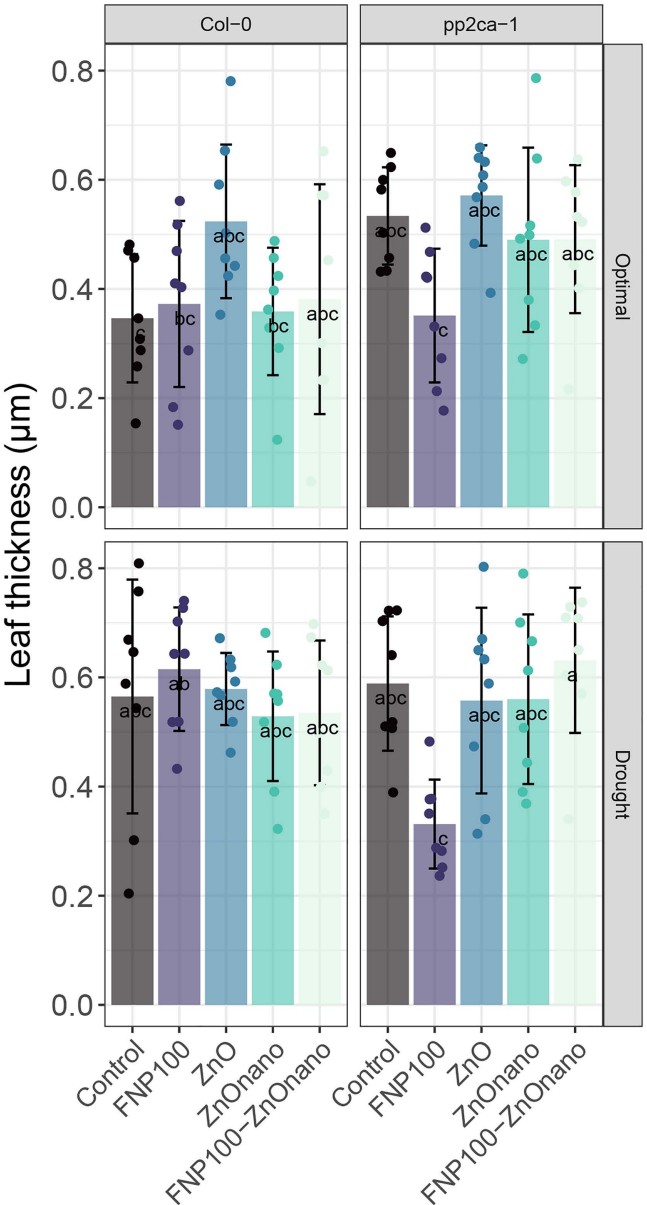

**Fig 6. Phenotypic changes are associated with both NP application and drought treatment.**

## Fullerenol and ZnO nano support photosynthetic assimilation and alleviate oxidative damage in drought-stressed plants

Multivariate analysis showed a clear separation between optimal and drought conditions. At the same time, among treatments, FNP100 exhibited the most substantial impact in treatments clustering (Fig 7), suggesting its profound impact on photosynthesis-related traits.

No statistically significant changes were observed in chlorophyll levels, based on SPAD levels, except for a reduction in pigment levels in ZnO-treated drought-exposed *pp2ca-1* plants (Fig 7). As expected, the photosynthetic net rate (PN) was reduced under drought conditions. However, this reduction in $CO_2$ assimilation was significantly alleviated by the FNP100 and FNP100-ZnO nano treatments, reaching the highest PN values under combined treatment. t. A similar pattern was observed for stomatal conductance (gs), which, in turn, was reflected in water use efficiency (WUE) values (Fig 7D, E). Interestingly, the opposite trend was seen in intrinsic water use efficiency (iWUE), where the same treatments caused a slight decrease compared to control plants.

Other photosynthetic-related traits, such as proton conductivity (gH+), steady-state proton flux (vH+), photosystem II quantum yield (Phi2), non-photochemical quenching (NPQt), and non-regulatory energy dissipation (PhiNO), did not show statistically significant changes compared to the corresponding controls (as detailed in the Supplementary Material, S2 Fig in S1 File). The only notable alteration was a significant reduction in NPQt in drought-exposed Col-0 plants treated with ZnO nano solutions (S2 Fig in S1 File).

The PCA plot showed a clustering of optimally grown plants for both genotypes on the right side of the plot, with FNP100 treatment being distinctly separated from the corresponding control groups. Under drought conditions, the FNP100 treatment also triggered a unique plant response in *pp2ca-1* plants, resulting in a migration farthest along both axes. In Col-0 drought-stressed plants, the ZnO nano treatment, followed by the combined FNP100-ZnO nano treatment and the individual FNP100 treatment, were separated entirely from the corresponding control towards the positive side of the Dim1 axis.

Significant changes in antioxidant enzyme activity and biochemical stress indicators were observed across the applied treatments, particularly those involving FNP (Fig 8). Multivariate analysis showed that FNP100 had the most substantial impact on enzyme activities and small molecules accumulation (Fig 8). Plant backgrounds clustered independently, as expected, with *pp2ca-1* mutant showing limited distinction within treatments.

The FNP100 treatment significantly reduced proline content in Col-0 plants exposed to drought, resulting in a six-fold decrease compared to the control (Fig 8F). The same trend was observed concerning ZnO nano and FNP100-ZnO nano combination treatment. Thus, proline levels were reduced compared to the control under drought conditions.

Reduced glutathione (GSH) content, which spiked sharply under water deficit compared to optimal conditions, was significantly lowered by both FNP100 and FNP100-ZnO nano treatments in both genotypes (significantly in Col-0) compared to the corresponding control.

For TBARS, a measure of lipid peroxidation, drought stress predictably led to a significant increase in lipid peroxidation levels, more so in Col-0. However, similar to GSH, FNP100, followed by the FNP100-ZnO nano and ZnO nano treatments, mitigated this increase, resulting in lower TBARS levels than control plants under drought conditions. This alleviation of oxidative stress was more pronounced in Col-0 plants.

Under optimal conditions, catalase activity was similar across all treatments, but it showed a marked increase in drought-stressed plants (Fig 8D). Compared to control plants under drought conditions, when observing Col-0 plants, catalase activity was further enhanced by the FNP100, followed by the ZnO nano and FNP100-ZnO nano joint treatments. In *pp2ca-1* mutant, this response was diametrical for FNP100 and FNP100-ZnO nano, with reduced catalase activities. A similar trend was observed for ascorbate peroxidase (APx) activity, with the FNP100 treatment causing a statistically significant increase in Col-0 compared to control plants under drought conditions.

The PCA plot revealed that in Col-0, the FNP100 treatment had the most substantial impact on antioxidant enzyme activity and stress indicators, as it was distinctly separated from other groups along the negative side of the Dim1 axis in

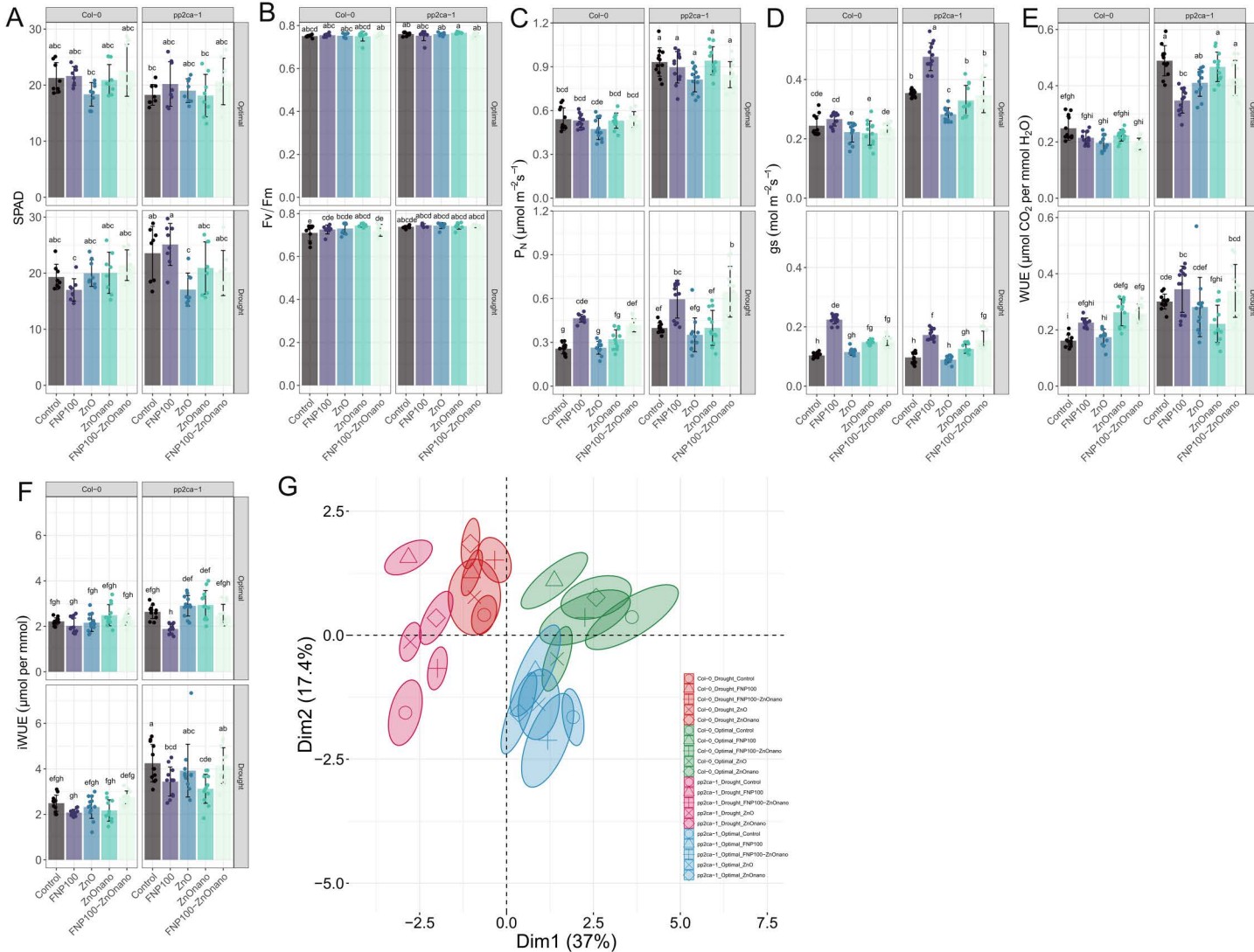

**Fig 7. The carbon sequestration capacity of NPs-treated plants is regulated at the stomatal level upon drought.** (A) Relative chlorophyll content. (B) Photochemical efficiency under light conditions (efficiency of PSII in converting absorbed light energy into chemical energy). (C) Net photosynthesis. (D) Stomatal conductance of water vapour. (E) Water use efficiency is expressed as the ratio of net photosynthesis and transpiration. (F) Intrinsic water use efficiency is expressed as the ratio of net photosynthesis and stomatal conductance of water vapour. (G) Principal component analysis plot. Data clustering is based on the treatments and plant background. Data are presented as mean±se, with independent measurements depicted as data points. Letters indicate significant differences based on the Tukey HSD test. The sample size is 15 replicates per condition. Detailed ANOVA output tables for each parameter are provided in the Supplementary Information S2 Table in S1 File.

drought. A notable separation was also observed for the FNP100-ZnO nano treatment, followed by the ZnO nano treatment. Interestingly, the bulk ZnO treatment showed minimal distinction from the control, with their plots almost completely overlapping, indicating a comparatively weaker effect. In optimal conditions, both genotypes agglomerated rather close on the plot. Observing *pp2ca-1* mutant plants in drought, the distinction is similar to that of Col-0 with FNP100, ZnO nano and FNP100-ZnO nano treatments clearly up on the plot towards the positive side of Dim1.

Environmental stress factors, especially drought, lead to excessive production of reactive oxygen species (ROS) and reactive nitrogen species (RNS), particularly in redox-active organelles such as chloroplasts, mitochondria, and peroxisomes.

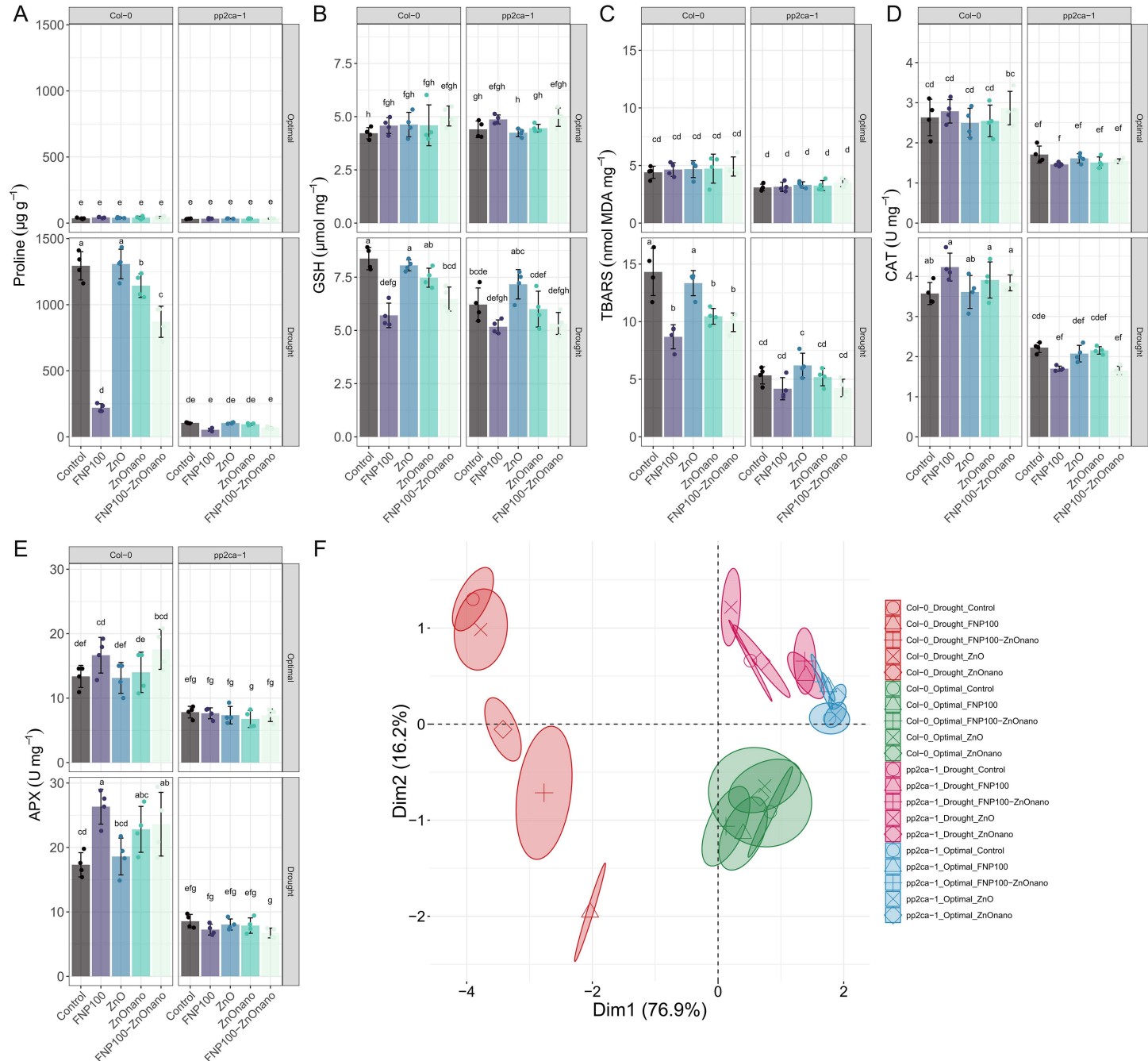

**Fig 8. Antioxidant system response and ROS scavenging are differentially regulated under FNP and Zn treatments.** (A) Accumulation of free proline. (B) Accumulation of reduced glutathione. (C) Thiobarbituric-reactive substances accumulation (TBARS). (D) Catalase activity. (E) Ascorbate-peroxidase activity. (F) Principal component analysis plot. Data clustering is based on the treatments and plant background. Data are presented as mean±se, with independent measurements depicted as data points. Letters indicate significant differences based on the Tukey HSD test. The sample size is four replicates per condition. Detailed ANOVA output tables for each parameter are provided in the Supplementary Information S3 Table in S1 File.

Here, intense electron flow generates ROS through mechanisms like the Fenton reaction or Haber–Weiss cycle [56]. When plants are exposed to various abiotic and biotic stresses, this increase in ROS production can damage vital biomolecules. However, ROS also serve as reliable signalling molecules that initiate acclimation responses [57]. Among these, hydrogen peroxide ($H_2O_2$), which is longer-lived than singlet oxygen ($^1O_2$), plays a critical role in the ROS signalling pathway, inducing strong stress responses [58]. The most reactive ROS in plant cells is the hydroxyl radical (OH•), derived from the superoxide anion radical ($O_2$•−) and $H_2O_2$, catalysed by iron, and capable of reacting with all biomolecules and metabolites.

Our results confirm that FNP treatments, both individually and in combination with ZnO nanoparticles, provide clear benefits in dampening ROS levels and stabilizing redox homeostasis in plant cells. We observed that shifts in catalase and ascorbate-peroxidase activities aligned with reductions in proline and glutathione spikes under drought stress, further supported by decreased TBARS levels, an oxidative stress indicator. Similar findings were previously reported in sugar beet, where fullerenol nanoparticles containing 24 hydroxyl (OH) groups demonstrated antioxidative potential as a foliar treatment [16]. Other studies have also validated the antioxidative effects of fullerenols ($C_{60}(OH)x$) in barley [10], spring wheat [13], *Brassica napus* [15,19] and *Solanum lycopersicum* [18].The direct antioxidant mechanisms of fullerenols have been characterized in different experimental setups. Fullerenols scavenge hydroxyl radicals via two pathways: firstly, hydroxyl radicals can add to the abundant sp² carbon atoms within fullerenol molecules [59,60] secondly, OH• radicals abstract hydrogen electrons from fullerenols, resulting in a relatively stable PHF radical [61]. Fullerenols' multiple hydroxyl (-OH) groups also enable the scavenging of OH• via a hydrogen abstraction mechanism. The electrophilic nature of fullerenols, ranging from 1 nm molecular structures to stable polyanionic particles (20–100 nm) over a wide pH range, categorizes them as exceptional antioxidants in biological systems [6]. Additionally, fullerenols neutralize $O_2$− and $^1O_2$ species via peroxyl radicals and hydrogen bonding [62,63] while also eliminating NO• radicals [8,64]. Studies have further validated fullerenols antioxidant properties in bacterial models [65].

Another mechanism of action involves an indirect effect of fullerenol on water supply. Fullerenols act as non-selective "sponges," noncovalently binding significant amounts of water. Although not experimentally confirmed in this study, previous work by our coauthors has shown that water molecules interlock within fullerenol self-assembled aggregates, with notable water release from solvation layers occurring when temperatures rise above 36°C [66]. This interlocked water may act as a reserve that can be gradually released when water potential drops under drought or temperature stress. Consistent with this, our findings demonstrate that fullerenol treatment under drought conditions reduces proline content, enhances stomatal conductance (more so under drought), and lowers intrinsic water use efficiency, with effects amplified in drought-sensitive mutants. Previous studies also observed proline reduction under fullerenol ($C_{60}(OH)_{24}$) treatment in drought-stressed sugar beet [16] and *Brassica napus*, where fullerenol application led to upregulation of aquaporin genes and significant increases in leaf relative water content and weight [19]. Notably, Ozfidan-Konakci et al. (2022) found that when pure $C_{60}$ (without OH groups and correlated hygroscopic properties) was applied to corn under cobalt stress, an antioxidant effect was observed, but proline concentrations increased, demonstrating a contrasting response [17].

These findings suggest that the combined antioxidant and the ability of FNPs to interact with water, attributable to their numerous surface hydroxyl (–OH) groups, might support increased photosynthetic rates, both when applied individually and with ZnO nanoparticles. The polarity of these hydroxyl groups facilitates hydrogen bonding with water molecules, potentially improving water availability and contributing eto enhanced stomatal conductance under drought stress in both Col-0 and *pp2ca-1* mutant plants.Although to a quantitatively smaller extent, a 10 mg/L foliar application of ZnO nanoparticles also enhanced Arabidopsis drought tolerance, reducing oxidative damage and supporting photosynthetic activity. Notably, ZnO nanoparticle treatment significantly reduced non-photochemical quenching (NPQt) in drought-stressed Col-0 plants (Fig S3 in S1 File). To the best of our knowledge, no studies have been conducted on the foliar application of ZnO nanoparticles specifically to *Arabidopsis thaliana*. However, the positive effects observed here align with previously described ZnO nano properties, including antioxidant activity and potential modifications to osmotic and carbohydrate metabolism and photosynthetic processes [23].

Regarding foliar applications to other plant species, recent studies confirm the beneficial impacts of ZnO nanoparticles at low concentrations (ranging from 25–200 mg/L). Similar to our findings, studies report positive effects of foliar ZnO nanoparticle applications on wheat (25–100 mg/L) [67], (50, 100, and 150 ppm) [68], (25–200 mg/L) [69], dragonhead (*Dracocephalum moldavica*, 40–400 mg/L) [70], foxtail millet (*Setaria italica*, 2.6 mg/L) [71], tomato (*Solanum lycopersicum*, 25–100 mg/L [72]; 75–125 mg/L [22]), rice (*Oryza sativa*, 60 and 120 mg/L [73]; 0.75–12 kg hm$^{-2}$ [74]), *Vicia faba* (50 and 100 mg/L) [75], lettuce (*Lactuca sativa*, 200 mg/L) [76], and eggplant (*Solanum melongena*, 50 and 100 mg/L) [77]. Across these species, ZnO nanoparticles at low concentrations supported antioxidative defense mechanisms, consistent with our Arabidopsis results.

However, studies on ZnO nanoparticle exposure to the root medium present variable outcomes, showing both stimulative and toxic effects. For instance, in *Brassica juncea*, beneficial effects were observed at ZnO nanoparticle concentrations ranging from 20–200 mg/L in soil applications [78], while *Linum usitatissimum* (1–1000 mg/L in MS agar medium) [79] and *Pleioblastus pygmaeus* (4–16 mg/L in MS agar medium) [80] exhibited boosted antioxidative responses. In contrast, higher ZnO nanoparticle concentrations generally led to growth inhibition and oxidative stress in several species. For instance, in tomato, concentrations of 400 and 800 mg/L in soil led to reduced growth, pigment content, and photosynthetic activity, whereas 200 mg/L did not affect these parameters [81]. Cytotoxic effects were reported in *Allium cepa* grown in hydroponic media at ZnO nanoparticle concentrations between 200–800 mg/L [82].

Interestingly, in *Arabidopsis thaliana*, most studies report toxicity from ZnO nanoparticle exposure when applied to the root medium rather than through foliar uptake. For example, Khan et al. (2019) observed no significant effects at 50 mg/L applied to the soil via watering, but toxic effects appeared at higher concentrations (100–300 mg/L) [83]. Similarly, Lee et al. (2009) reported toxicity in Arabidopsis grown in agar medium at ZnO nanoparticle concentrations of 400, 2000, and 4000 mg/L, attributing the increased uptake rate to ZnO's 33-fold higher solubility compared to standard zinc salts like $ZnCl_2$ [84]. ZnO nanoparticle size was also a determinant, with nanoparticles showing more significant toxicity than micron-sized particles at equivalent concentrations. In hydroponic media, even low ZnO nanoparticle concentrations (0.16–100 mg/L) negatively impacted Arabidopsis growth, increasing abscisic acid levels and decreasing growth-promoting hormones, leading to an overall inhibitory response [85]. Additionally, a seven-day ZnO nanoparticle exposure at 100 mg/L induced 660 genes and inhibited 826 genes, including those responsible for cellular organization, biogenesis, and plant defence [86]. These findings indicate that the effects of ZnO nanoparticles on plants depend heavily on concentration, plant species, treatment type (foliar vs. root), and nanoparticle size, collectively influencing ZnO bioavailability and ion release. A general trend suggests that single, low-concentration foliar applications below 200 mg/L often yield beneficial outcomes for plant growth and stress resistance, consistent with our findings in *Arabidopsis thaliana*.

Additionally, to the best of our knowledge, this is the first study to investigate the synergistic action of fullerenol and ZnO nanoparticles applied foliarly to mitigate drought stress, thereby opening a novel direction for nano-enabled plant resilience strategies. It is essential to emphasize that the FNP-ZnO nano combined treatment revealed a synergistic effect driven by the complementary mechanisms of both applied nanoparticles. While FNP exhibits strong hygroscopic and antioxidant properties, enabling the modulation of cellular water potential and enhancing ROS scavenging under stress conditions [6,66], ZnO nanoparticles, on the other hand, may serve as sources of micronutrients and redox modulators, influencing osmotic balance, ion homeostasis, and oxidative signaling [23,68,69]. The superior performances were evident in boosting photosynthetic performance (Fig 7B, D) and minimizing oxidative damage under the combined treatment (Fig 8B, C, E), suggesting an additive modulation of key drought-related pathways.

## Drought response gene expression is modulated in response to FNP and ZnO nano

Plants possess an adaptive robustness in response to osmotic stresses, such as drought. Aiming to bridge nanomaterials usage in the environment, it is necessary to untangle the mechanisms underlying the beneficial effects of FNP and ZnO nanoparticles in plant resilience and tolerance to water shortage. Overall, the unique physicochemical properties of ENPs,

such as a small size, a large surface area, high cation exchange capacity (CEC), high absorbability, and potent free radical scavenging capacity, ultimately contribute to the plant's ability to withstand drought stress [1,4]. It was previously reported that nanoparticle applications, including FNP and Zn, can modify the transcription of genes involved in biosynthesis and signal transduction of various plant hormones in the first-line ABA [15,87]. ABA is a critical stress-signaling hormone, one of the core components that can trigger a complex network of signal transduction cascades in molecular events, leading to physiological and biochemical modifications and consequently contributing to enhanced drought resistance by diminishing osmotic stress [1,88] Recently, Ning et al. (2024) observed that fullerenol enhances osmotic stress tolerance by modulating antioxidant transcriptional and epigenetic processes, impacting environmental signaling pathways [89]. The authors concluded that fullerenol conferred alleviation of osmotic stress by triggering the expression of antioxidant genes, altering the transcription level and chromatin accessibility of specific genes involved in mitigating osmotic stress, and modifying chromatin structure. These findings indicate that FNP mediates changes in both early and late responses to osmotic stress.

We aimed to analyze the ABA signaling network associated with drought stress responses to enhance our understanding of how FNP and ZnO nanoparticles affect the expression of stress-responsive genes triggered by water loss. Fig 9 presents the expression of selected drought stress-related genes for both Col-0 and the *pp2ca-1* mutant. Based on multivariate analysis, it has been shown that different water regimes tested (optimal vs water shortage) segregate distinctly in Col-0 background. However, this observation was only present in the control group in the *pp2ca-1* mutant. It could be inferred that NP presence in planta triggers osmotic adjustment overall, resulting in increased ABA-related gene transcript levels, which, without PP2CA regulation, leads to gene up-regulation independent of drought presence/absence. Fullerenol-induced modulation of ABA metabolism was previously reported by Xiong et al. (2018), who demonstrated that under drought stress, the application of 1 mg L$^{-1}$ fullerenol significantly upregulated the expression of NCED3, a key gene in ABA biosynthesis, while downregulating CYP707A3, which is involved in ABA catabolism, in *Brassica napus* [15]. These results suggest that FNP's impact on plant metabolism is partly mediated through ABA-regulatory signaling cascades.

PYR/PYL/RCAR proteins, acting as ABA receptors, group A- protein phosphatase 2C (PP2C) as a strong negative regulator of ABA signal transduction, and SnRK2s as positive regulators are core components in ABA signaling, possessing a putative role in enhancing plants' resistance to drought [30]. For instance, PYL4 overexpression enhances drought resistance in Arabidopsis [90]. However, we reported down-expression of PYL4 in Col-0 plants under drought, regardless of the applied treatment (Fig 9A). On the contrary, FNP provoked overexpression of PYL4 in *pp2ca-1* mutant in both optimal and drought treatments. Similarly, PYR1 (Fig 9B) showed higher expression in mutant plants than in Col-0 plants, and combined FNP100-ZnO nano treatment induced a significant up-regulation under optimum conditions, indicating that combined treatment elicited more robust responses than either nanoparticle alone. Besides, we can see that the FNP effect is more pronounced in plants hypersensitive to ABA, potentially indicating that FNP primed the plants to a more robust stress response. Simultaneously, PP2CA gene expression in Col-0 plants was upregulated in both conditions (control and drought) regardless of the applied treatment. At the same time, it was down-regulated in the *pp2ca-1* mutant, demonstrating partial silencing of PP2CA gene activity in ABA hypersensitivity.

Regarding PP2CA, a negative regulator of ABA signaling, FNP, and ZnO nano treatments provoked decreased PP2CA expression under drought stress in Col-0 plants (Fig 9C). On the contrary, In the pp2ca-1 mutant, as expected, PP2CA expression was significantly reduced, confirming the absence of the functional gene in this background. The down-regulation of PP2CA in response to drought was particularly pronounced in treatments with FNP100 and ZnO nano. These contradictory expression patterns can be explained by introducing the negative feedback regulatory mechanism, which modulates the initial ABA response; therefore, ABA-mediate up-regulation of PP2Cs genes simultaneously with the downregulation of PYLs genes [87]. Additionally, monomeric PYL genes, including PYL4, interact with PP2Cs in an ABA-independent manner and can interact with PP2Cs even in the absence of ABA [91]. Similarly, the expression level of ABI2, another protein phosphatase 2C, which acts as a negative regulator of ABA signaling, was suppressed by NPs

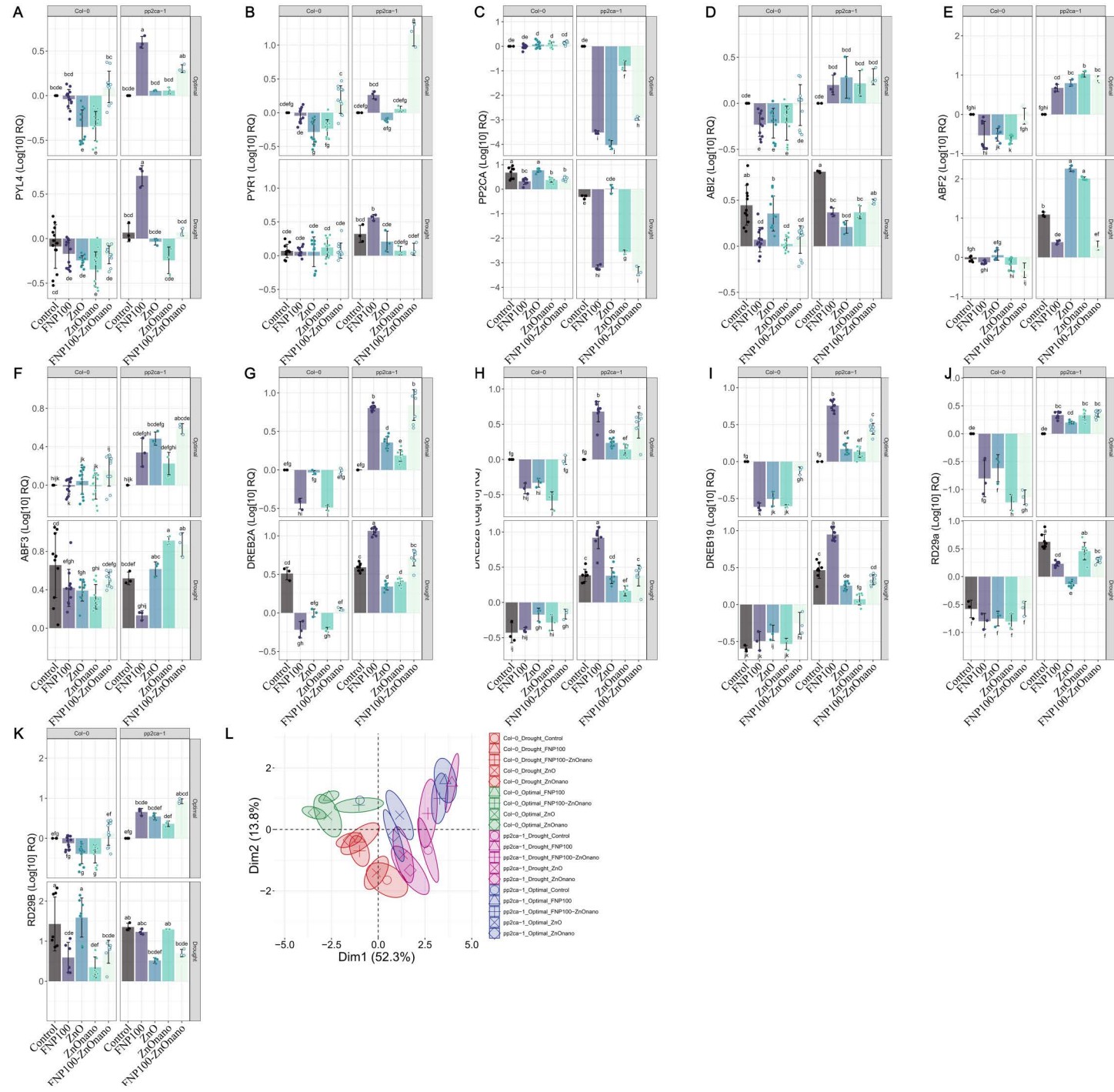

**Fig 9. Relative gene expressions and PCA plots of analyzed drought–related genes.** Data are presented as mean±se, with independent measurements depicted as data points. Letters indicate significant differences based on the Tukey HSD test. Detailed ANOVA output tables for each parameter are provided in the Supplementary Information Table S4 in S1 File.

treatments under drought stress in both Col-0 and *pp2ca-1* mutant (Fig 9D).These molecular responses, including the downregulation of PP2CA and ABI2 under drought conditions, are reflected in physiological phenotypes such as increased stomatal conductance and improved $CO_2$ assimilation (Fig 7). This suggests that modulation of ABA signaling plays a key role in enhancing drought resilience, as evidenced by the observed phenotypic outcomes. AREB1/ABF2, AREB2/ABF4, and ABF3 are transcription factors, that act as downstream mediators of ABA stress response. They are highly induced by osmotic stress and are thus involved in drought stress tolerance [92,93]. We observed overexpression of ABF2 and ABF3 (Fig 9E, F); however, in both Col-0 and *pp2ca-1* mutant plants under drought conditions, FNP and FNP100-ZnO nano treatments provoked a decrease in ABF2 and ABF3 expression, where a more pronounced effect was evident in mutant plants. In line with that, Xiong and Ma (2022) reported that the foliar application of fullerenols mitigates the negative effect of drought at the transcription level [19]. Based on GO analysis, the authors observed that in drought conditions, fullerenol leads to the down-regulation of genes related to protein kinase activity, including SnRKs. Since ABFs are activated via ABA-dependent phosphorylation regulated by SnRK2s, our finding implies that FNP triggers the ABA signaling network and modifies the response of transcription factors that regulate gene expression downstream of SnRKs. However, since those transcription factors are not exclusive components of the ABA signaling, their expression overlaps with genes involved in ABA-independent pathways.

DREB genes encode transcription factors binding to dehydration-responsive elements (DREs) and promote drought-responsive genes independent of the ABA pathway. Similar expression pattern, with few minor exceptions, was established for all three analyzed DREB transcription factors, DREB2A, DREB2B, and DREB19, which show constitutive upregulation in mutant plants (Fig 9G, H, I). Under optimal conditions, all treatments decreased the expression levels of DREB2A, DREB2B, and DREB19 in Col-0 plants. Conversely, in *pp2ca-1* mutant, all treatments significantly upregulated DREB genes, particularly in both FNP treatments. Under drought, significant changes in DREBs expression occurred between Col-0 and *pp2ca-1* plants. In Col-0 all DREBs were downregulated regardless of treatment, while in *pp2ca-1*, the opposite trend was observed, with the most potent effect in FNP treatments. This suggests a consistent impact of treatments on these ABA-independent genes during drought stress. We demonstrated that FNP triggers DREBs TF by inducing their overexpression, which is noticed in mutant plants, while this effect is suppressed in FNP100-ZnO nano. Furthermore, the upregulation of DREB genes in FNP-treated ABA-hypersensitive mutants under drought stress corresponds with the reduction of oxidative stress markers and an increase in antioxidative enzyme activity (Fig 8), indicating its functional role in ROS scavenging. Indeed, it can be seen that FNP is engaged in ABA signal transduction, but we still cannot clearly distinguish its involvement in ABA-dependent vs. independent pathways. Although it was previously stated that FNP mediates changes in both early and late responses to osmotic stress [89], time-course experiments would provide a more detailed mechanistic resolution of gene regulation under drought and FNP and ZnO-nano treatments.

RD29A and RD29B are highly induced by various types of abiotic stress and are regulated primarily by ABA-dependent pathways. However, RD29A is also influenced by ABA-independent signaling mediated by DREB/CBF transcription factors. In our data, RD29A showed a partially overlapping expression trend with DREB genes, suggesting potential co-regulation, whereas RD29B exhibited a response more consistent with classical ABA-dependent signaling genes (Fig 9J, K).

Under optimal conditions, RD29A expression was downregulated in Col-0 but upregulated in *pp2ca-1* mutant across all treatments. Under drought, in *pp2ca-1* mutant, RD29A expressions were significantly suppressed by all treatments except for ZnO nano, where changes were not significant. ABA-inducible RD29B expression showed insignificant variation in Col-0 under optimal conditions, while it was upregulated under drought but suppressed by FNP100 and ZnO nano treatments, both individually and combined. In *pp2ca-1* mutants, a similar pattern was observed, although the changes were not statistically significant.

The PCA plot (Fig 9L) demonstrated a clear separation between the Col-0 and *pp2ca-1* groups while overlapping among treatments was evident in *pp2ca-1* mutant plants. Under optimal conditions, PCA analysis revealed treatment-induced shifts in gene expression in both Col-0 and *pp2ca-1* mutants. FNP100 and FNP100-ZnO nano treatments

caused the most pronounced shift along Dim1, suggesting a significant impact on drought-related genes. Under drought, the separation between control and treated groups became more pronounced, especially for FNP100 and ZnO nano treatments. In the Col-0 background, ZnO-treated plants showed the most robust response to water deficit, while FNP100, FNP100-ZnO nano, and ZnO nano treatments suppressed the drought-induced gene expression shift. In *pp2ca-1* mutants under drought, the FNP100 treatment caused the most considerable shift along Dim1, with the FNP100-ZnO nano treatment showing a similar tendency. The ZnO nano and ZnO treatments are clustered on the opposite side of Dim1.

To summarize, FNP included an increased upregulation of most tested genes in pp2ca-1 mutant plants under optimal conditions, suggesting that fine-tuning, to a certain extent, is present and initiated by FNP, which stands out in hypersensitive plants. A putative assumption is that FNP, through previously characterized sponge hygroscopic effect, creates a mild low-magnitude osmotic disturbance, which initiates and prepares the cell to stress. Such mild initial osmotic disturbance is not significant in provoking shifts in antioxidative and biochemical stress indicators in wild type, while we observed that in ABA hypersensitive *pp2ca-1* mutant in optimal conditions. Thus, it is possible that FNP priming establishes a hormesis effect and, in drought conditions, causes shifting in signaling cascades by regulating TFs and stress-related genes, which is indirectly linked also to its antioxidant effect. It is still directly unconfirmed whether additional interlocked water depots are steadily formed in the plant cell environment in FNP presence and whether they further contribute to drought acclimation. Based on these single gene expressions, we cannot clearly define a time course of the initial and most efficient FNP effect in response to water scarcity; thus, further studies are required to evaluate these events and expand our knowledge of the action mechanisms. Even though the results of this study are elucidated on *Arabidopsis thaliana*, which served as an effective model for elucidating the physiological and molecular mechanisms of nanoparticle action, further studies are needed to validate these findings in crop species under agronomically relevant conditions. Such translational research is essential for determining the practical applicability of nanoparticle-based stress mitigation strategies. Although the beneficial effects of ZnO nanoparticles, fullerenol, and their combination have been demonstrated under controlled conditions, scaling these approaches to field-level agriculture requires addressing challenges such as application methods, formulation stability, cost-effectiveness, environmental persistence, and potential trophic transfer risks. Additionally, future research should assess long-term ecological impacts and consider regulatory frameworks to ensure their safe and sustainable use in commercial crop production. Moreover, considering the current limitation in understanding how nanoparticles trigger both ABA-dependent and ABA-independent stress signaling pathways, additional mechanistic studies are necessary to clarify these processes and evaluate the role of nanoparticles in other hormonal pathways.

## Conclusions

This study demonstrates for the first time that foliar application of low-dose ZnO nanoparticles (10 mg/L), alone or in combination with fullerenol nanoparticles (FNP), enhances drought tolerance in *Arabidopsis thaliana*. The observed improvements in redox balance, photosynthetic performance, and stomatal regulation point to multiple physiological pathways to support drought acclimation. At the molecular level, distinct gene expression patterns suggest involvement of both ABA-dependent and independent pathways. Notably, the synergistic effect of the combined application underscores the potential of nanoparticle-based biostimulants in plant stress management. These findings offer a promising direction for future research in nanotechnology-driven strategies for sustainable agriculture under climate stress.

## Supporting information

**S1 File. Supplemental Information.** Analyzed drought related genes and their sequences, Supplemental Figures (S2-S3). **Supplemental Tables S2-S6**. Mixed-model ANOVA outputs for different analyzed datasets.
(ZIP)

## Author contributions

**Conceptualization:** Milan Borišev, Aleksandar Djordjević.

**Data curation:** Ana Joksimović, Danijela Arsenov, Milan Borišev, Milan Zupunski, Ivana Borisev.

**Formal analysis:** Ana Joksimović, Milan Zupunski.

**Investigation:** Ana Joksimović, Danijela Arsenov, Milan Borišev, Milan Zupunski, Ivana Borisev.

**Methodology:** Ana Joksimović, Danijela Arsenov, Milan Borišev, Milan Zupunski, Ivana Borisev.

**Resources:** Aleksandar Djordjević, Milan Zupunski.

**Supervision:** Aleksandar Djordjević, Ivana Borisev.

**Validation:** Aleksandar Djordjević, Milan Zupunski, Ivana Borisev.

**Visualization:** Milan Zupunski.

**Writing – original draft:** Ana Joksimović, Danijela Arsenov, Milan Borišev, Milan Zupunski, Ivana Borisev.

**Writing – review & editing:** Danijela Arsenov, Milan Borišev, Aleksandar Djordjević, Milan Zupunski, Ivana Borisev.

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
