## [Decision Letter · Decision Letter 0]

16 May 2025

Dear Dr. Zupunski,

Thank you for submitting your manuscript to PLOS ONE. After careful consideration, we feel that it has merit but does not fully meet PLOS ONE’s publication criteria as it currently stands. Therefore, we invite you to submit a revised version of the manuscript that addresses the points raised during the review process.

We look forward to receiving your revised manuscript.

Kind regards,

Ajit Prakash, PhD

Academic Editor

PLOS ONE

Journal Requirements:

“The authors gratefully acknowledge the financial support of the Ministry of Science, Technological Development and Innovation of the Republic of Serbia (Grants No. 451 -03-137/2025-03/ 200125 & 451- 03-136/2025-03/ 200125), and the DEAL initiative HHU (M.Ž.).”

“The authors gratefully acknowledge the financial support of the Ministry of Science, Technological Development and Innovation of the Republic of Serbia (Grants No. 451 -03-137/2025-03/ 200125 & 451- 03-136/2025-03/ 200125), and the DEAL initiative HHU (M.Ž.).”

“The author(s) received no specific funding for this work”

5. We notice that your supplementary figures are uploaded with the file type 'Figure'. Please amend the file type to 'Supporting Information'. Please ensure that each Supporting Information file has a legend listed in the manuscript after the references list.

Reviewers' comments:

Reviewer's Responses to Questions

**Comments to the Author**

1. Is the manuscript technically sound, and do the data support the conclusions?

Reviewer #1: Yes

Reviewer #2: Yes

Reviewer #3: Yes

Reviewer #4: Partly

2. Has the statistical analysis been performed appropriately and rigorously?

Reviewer #1: Yes

Reviewer #2: Yes

Reviewer #3: Yes

Reviewer #4: No

3. Have the authors made all data underlying the findings in their manuscript fully available?

Reviewer #1: Yes

Reviewer #2: Yes

Reviewer #3: Yes

Reviewer #4: No

4. Is the manuscript presented in an intelligible fashion and written in standard English?

Reviewer #1: Yes

Reviewer #2: Yes

Reviewer #3: Yes

Reviewer #4: Yes

Reviewer #1: Summary of Major Comments (Overall Paper):

Incorporating feedback from all sections, you could include points like:

Clarification and strengthening of key mechanisms: Ensure the discussion addresses specific physiological and molecular pathways more clearly, with emphasis on how the nanoparticles' physicochemical properties impact drought stress at the cellular and molecular levels.

Further exploration of synergy: Given that the combination of FNP and ZnO nano has a synergistic effect, ensure that this interaction is clearly explained in both the results and discussion sections.

Experimental considerations: If there are limitations in experimental design (e.g., the use of a single model plant like Arabidopsis), these should be discussed, and potential extensions to other crops or field studies could be suggested.

Suggestions for scalability and practical applications: Providing a more explicit connection between the findings and real-world applications, including challenges to scalability, would improve the practical relevance of the paper.

Acknowledgment of unconfirmed mechanisms: Future studies to better define the mechanisms and explore other potential interactions (like the role of antioxidant properties) should be clearly outlined.

Statistical rigor and data transparency: Ensure statistical methods and data representation are transparent and appropriately described, as this strengthens the reliability of the conclusions.

Reviewer #2: The current MS entitled “ Foliar application of fullerenol and zinc oxide nanoparticles improves stress resilience in drought- sensitive Arabidopsis thaliana”. In this sense, manuscript's is appropriate in order to represents the the subject of the research. The abstract of the manuscript is precise and present whole manuscript objectives. The references used in the manuscript are up- to-date ,satisfactory and convenient. The conclusion should only consist of key findings and significance.

Reviewer #3: Comments

The manuscript “Foliar application of fullerenol and zinc oxide nanoparticles improves stress resilience in drought-sensitive Arabidopsis thaliana” presents a detailed and thorough study about characterization and chemical properties of fullerenol nanoparticles (FNP) and zinc oxide nanoparticles (ZnO nano). Arabidopsis thaliana is an interesting choice to undertake these studies as it is regarded as model plant to study biology, stress response and genetics. As a result of nanoparticle treatment, study about changes in gene expression emphasized synergistic impact of combining ZnO nanoparticles with FNPs. However, a few queries need to be addressed to make this manuscript more rigorous.

1. Hydroxylated fullerenes have already been studied extensively for foliar applications previously. What leads to choice of fullerenol in your research?

2. In introduction, are ENPs, engineered nanoparticles? Explain a little about ENPs to make the readers better understand them.

3. What is the rationale behind choosing fullerenes with 24 hydroxyl groups? As 18 to 40 hydroxyl groups is considered the practical range for most applications, why did the study not consider fullerenols with different numbers of functional groups?

4. Under the heading of Chemicals Used, synthesis of fullerenol NPs has not been explained. It is not appropriate to present it as a reference.

5. In topic, RNA Isolation and quantitative RT-PCR analysis, how the RNA quality and purity ensured?

6. The results of SEM, DLS, Zeta potential, XRD Diffractograms, and FTIR of the FNP100-ZnO nano combination effectively characterize their formation.

7. During experiment were TBARS levels correlated statistically with non-enzymatic antioxidants like proline/glutathione?

8. Effects of different ZnO nanoparticle concentrations on plant growth and stress resistance are reviewed extensively, quoting numerous references. It looks like little overemphasized in comparison to FNP100-ZnO nano combination.

9. What is the rationale behind excluding any other plant hormone pathways, like auxin or salicylic acid for studying the effects of nanoparticles on plant stress responses?

Reviewer #4: This manuscript explores the physiological, biochemical, and molecular effects of foliar application of fullerenol nanoparticles (FNP) and zinc oxide nanoparticles (ZnO NPs), alone and in combination, on drought-stressed Arabidopsis thaliana (Col-0 and pp2ca-1 mutant). The study suggests that both nanoparticles, especially FNP, enhance drought tolerance by modulating antioxidant enzyme activities, gene expression related to ABA signaling, and water use efficiency. The manuscript has strong potential but requires major revisions to address critical issues related to mechanistic interpretation and statistical clarity. I recommend a major revision and encourage the authors to significantly enhance the methodological clarity and depth of the discussion.

1. Gene expression results are not convincingly linked to phenotype. The expression data for ABA-dependent and independent pathways are descriptive but lack a direct connection to the observed physiological effects. Time-course experiments would provide a clearer mechanistic understanding of gene regulation under drought and NP treatment.

2. Nanoparticles can accumulate or trigger unintended responses. No analysis is provided for potential phytotoxicity, metal accumulation, or long-term side effects in the plants. Authors must include at least a discussion of possible NP toxicity or ecological implications.

3. While the discussion is extensive, it often summarizes results without offering mechanistic insights. How exactly FNPs enhance water retention or trigger ABA pathways.

4. “FNPs act as water sponges and gradually release water under drought...” This hypothesis lacks experimental confirmation in the current study. No direct measurement of water retention or release by FNPs in planta is presented.

5. “RD29A exhibited expression patterns more closely aligned with those of DREB genes...” The gene expression data do not fully support this equivalence. Provide statistical support or rephrase more cautiously.

6. Hygroscopic properties of fullerenol with no water retention data. Conduct thermogravimetric analysis or remove claim.

7. Missing RIN values and amplification efficiency data (violates MIQE guidelines).

8. Manuscript lacks accessible raw data for reproducibility. Include supplementary tables with raw data.

9. Statistical reporting is incomplete (missing F-values, degrees of freedom, p-values).

10. Use consistent terminology throughout: sometimes ZnO nano, other times ZnO NPs or ZnO nanoparticles.

11. ENPs is undefined upon first use. Expand to engineered nanoparticles (ENPs) initially.

**Do you want your identity to be public for this peer review?** For information about this choice, including consent withdrawal, please see our Privacy Policy

Reviewer #1: **Yes: ** Dhakshnamoorthy Vellingiri

Reviewer #2: **Yes: ** Anjali

Reviewer #3: **Yes: ** Harpreet Kaur

Reviewer #4: No

---

## [Author Response · Author response to Decision Letter 1]

30 Jun 2025

We sincerely thank the editor and reviewers for their constructive comments and thoughtful suggestions, which have significantly contributed to the improvement of our manuscript's quality. We believe that the revised version now meets the standards and publication criteria of PLOS ONE. All suggested revisions have been carefully addressed, and the updated manuscript includes a marked-up version with all changes highlighted. A point-by-point response to comments is available in "Response to reviewers.pdf".

On behalf of all authors,

Dr. Milan Zupunski

---

## [Decision Letter · Decision Letter 1]

24 Jul 2025

Foliar application of fullerenol and zinc oxide nanoparticles improves stress resilience in drought-sensitive Arabidopsis thaliana

PONE-D-25-20535R1

Dear Dr. Zupunski,

We’re pleased to inform you that your manuscript has been judged scientifically suitable for publication and will be formally accepted for publication once it meets all outstanding technical requirements.

Kind regards,

Ajit Prakash, PhD

Academic Editor

PLOS ONE

Additional Editor Comments (optional):

Reviewers' comments:

Reviewer's Responses to Questions

**Comments to the Author**

Reviewer #1: All comments have been addressed

Reviewer #2: All comments have been addressed

Reviewer #3: All comments have been addressed

Reviewer #4: All comments have been addressed

2. Is the manuscript technically sound, and do the data support the conclusions?

Reviewer #1: Yes

Reviewer #2: Yes

Reviewer #3: Yes

Reviewer #4: Yes

3. Has the statistical analysis been performed appropriately and rigorously?

Reviewer #1: Yes

Reviewer #2: Yes

Reviewer #3: Yes

Reviewer #4: Yes

4. Have the authors made all data underlying the findings in their manuscript fully available?

Reviewer #1: Yes

Reviewer #2: Yes

Reviewer #3: Yes

Reviewer #4: Yes

5. Is the manuscript presented in an intelligible fashion and written in standard English?

Reviewer #1: Yes

Reviewer #2: Yes

Reviewer #3: Yes

Reviewer #4: Yes

Reviewer #1: This study presents a comprehensive investigation into the effects of fullerenol (FNP) and zinc oxide (ZnO) nanoparticles on Arabidopsis thaliana under drought stress, offering valuable insights into their potential for enhancing plant resilience. The nanoparticle characterization is largely thorough, though minor clarifications are needed regarding DLS "referent materials" and ensuring consistency between SEM images and text. The experimental design is robust, utilizing both wild-type and a drought-hypersensitive mutant, which strengthens the interpretation of molecular mechanisms.

The results consistently demonstrate the biostimulatory and stress-alleviating effects of FNP, largely attributed to its unique antioxidant and hypothesized hygroscopic properties. ZnO nanoparticles also show beneficial effects, particularly in reducing NPQt. A key finding is the strong indication of synergistic protective effects when FNP and ZnO nano are combined, especially in enhancing photosynthetic performance and mitigating oxidative damage.

However, the discussion of gene expression data, particularly the PCA interpretation, needs further clarity and direct textual support for some inferences. The observed downregulation of stress-responsive genes like ABF2/3 with nanoparticle treatment under drought is a highly significant and positive finding, as it suggests a successful mitigation of the stress response at the molecular level, and this point could be emphasized more clearly.

Overall, this research is a strong contribution to the field of nano-enabled agriculture, providing compelling evidence for the potential of FNP and ZnO nanoparticles, especially in combination, to improve drought tolerance in plants.

Reviewer #2: The manuscript titled "Foliar application of fullerenol and zinc oxide nanoparticles improves stress resilience in drought-sensitive Arabidopsis thaliana" presents a comprehensive and insightful study .Data representation, statistical methods, and data rigor are all appropriately and clearly explained. In the conclusion, the authors highlight the key findings and their scientific importance. They've also thoughtfully addressed all the questions raised by other reviewers.

Reviewer #3: (No Response)

Reviewer #4: I have carefully evaluated the revised manuscript entitled "Foliar application of fullerenol and zinc oxide nanoparticles improves stress resilience in drought-sensitive Arabidopsis thaliana". The authors have thoroughly addressed all of my previous comments and provided satisfactory responses. The revised version significantly improves the clarity, scientific rigor, and overall presentation of the study. I recommend the manuscript for acceptance and publication in PLOS ONE.

**Do you want your identity to be public for this peer review?** For information about this choice, including consent withdrawal, please see our Privacy Policy

Reviewer #1: **Yes: ** Dhakshnamoorthy Vellingiri

Reviewer #2: **Yes: ** Anjali Das

Reviewer #3: **Yes: ** Harpreet Kaur

Reviewer #4: No

---

## [Editor Report · Acceptance letter]

PONE-D-25-20535R1

PLOS ONE

Dear Dr. Zupunski,

I'm pleased to inform you that your manuscript has been deemed suitable for publication in PLOS ONE. Congratulations! Your manuscript is now being handed over to our production team.

Kind regards,

on behalf of

Dr. Ajit Prakash

Academic Editor

PLOS ONE